# Couple Conflict and Intimate Partner Violence during the Early Lockdown of the Pandemic: The Good, the Bad, or Is It Just the Same in a North Carolina, Low-Resource Population?

**DOI:** 10.3390/ijerph19052608

**Published:** 2022-02-24

**Authors:** Jennifer Langhinrichsen-Rohling, Grace E. Schroeder, Ryan A. Langhinrichsen-Rohling, Annelise Mennicke, Yu-Jay Harris, Sharon Sullivan, Glori Gray, Robert J. Cramer

**Affiliations:** 1Department of Psychological Science, University of North Carolina at Charlotte, Charlotte, NC 28223, USA; gschroe1@uncc.edu; 2Department of Psychology, Emory University, Atlanta, GA 30322, USA; rlanghi@emory.edu; 3School of Social Work, University of North Carolina at Charlotte, Charlotte, NC 28223, USA; amennick@uncc.edu; 4Psychology for All, Charlotte, NC 28227, USA; yuharris@gmail.com (Y.-J.H.); sullidsl@bellsouth.net (S.S.); drgray@centralpsychcharlotte.com (G.G.); 5Department of Public Health Sciences, University of North Carolina at Charlotte, Charlotte, NC 28223, USA; rcramer4@uncc.edu

**Keywords:** couple conflict, intimate partner violence, COVID-19, pandemic

## Abstract

The COVID-19 pandemic has forced couples to navigate illness-related stressors and unique public health responses, including extended lockdowns. This study focused on under-resourced North Carolina residents (*n* = 107) who self-reported changes in relationship conflict (Increased, Decreased, Stayed the Same) and intimate partner violence (IPV) during the pandemic. We expected high rates of increased conflict and IPV since the start of the pandemic. We then sought to determine the associations between dyadic changes in conflict and reports of IPV and pandemic-related experiences and responses. Participants completed a brief online survey assessing their demographics, COVID-19 exposure/stressors, and pandemic responses. As expected, reports of increased couple conflict were related to difficulties getting needed social support, loss of health insurance, more fear and worry, stress, pain, and greater use of alcohol and/or illicit drugs, related to the coronavirus. Participants reporting increased conflict were also more likely to be unemployed. Conversely, reports of decreased conflict were associated with being ill from the virus (48.9%), having health insurance, and working part time. Substantial amounts of IPV were reported (62.2% of the sample); however, increased conflict and IPV were unrelated. Those reporting No IPV were less likely to be receiving public assistance but more likely to have home responsibilities due to the virus. They also reported increased social interactions and less use of alcohol than those reporting IPV perpetration. Findings highlight key associations among pandemic experiences and responses, IPV, and couple functioning in an under-resourced sample. Efforts to facilitate coping, resilience, and tolerating uncertainty may facilitate cooperative and safe couple functioning throughout the pandemic.

## 1. Introduction

The long-lasting and evolving COVID-19 pandemic has impacted much of the world, leading to millions of deaths, thousands of jobs lost, and extended quarantines during which many people were isolated in their homes, often spending unexpected and extended time with their significant others [1,2]. During the early stages of the pandemic, when widespread lockdowns were common, some began to wonder about the unintended and potentially negative impacts of this public health strategy [3]. Consistent with this concern, both the pandemic itself, as well as efforts to prevent the spread of the pandemic, have been associated with social, economic, and health/mental health difficulties across the United States, with pandemic-related effects constituting a substantial stressor for many individuals [4,5]. The pandemic and corresponding safety efforts have also put stress on relationships and families. Specifically, recent research indicates that, for many, experiencing the pandemic in general, and lockdown in particular, has been associated with increased feelings of loneliness and isolation, as well as reduced access to social support outside of the home [6,7,8,9].

These findings underscore the importance of understanding the functioning of intimate relationships during the pandemic. Theoretically, intimate relationships could function in two disparate ways. On the one hand, it is possible that pre-existing couple relationships could act as a source of support during COVID-19, helping to mitigate the negative impacts associated with isolation, uncertainty, and ill health. Conversely, if conflict were to increase in these romantic relationships, either due to the pandemic and its health, social, and economic impacts, or as a consequence of quarantining together, it could have adverse implications for the health and welfare of individuals. Of note, one particularly problematic outcome related to increased couple conflict could be rising rates of intimate partner violence. Some have even suggested that increasing rates of intimate partner violence during the pandemic, as demonstrated by increased calls to police and/or domestic violence hotlines in some communities [10] and among studies emerging internationally [11], constitutes a second, unexpected, public health pandemic. For a narrative review of increased rates of IPV during the pandemic and potential associated reasons for this uptick, consider Moreira and Pinto da Costa (2020) [12].

Numerous pre-pandemic studies have demonstrated that one of the strongest predictors of intimate partner violence is intense relationship conflict [13,14,15]. Additionally many recent studies indicate that the COVID-19 pandemic has resulted in increased rates of intimate partner conflict [16,17]. For example, in a study of Belgian adults recruited via an online survey, Schokkenbroek et al. (2021) assessed five aspects of respondent’s intimate relationships during the pandemic: increased conflict; diverging attitudes regarding relationship and life; restrictions; less connectedness; and neglect by partner. Although both men and women reported experiencing a variety of dyadic symptoms, the most widely endorsed pandemically-related impact was increased relationship conflict.

Given that an increase in conflict in romantic relationships has been robustly associated with intimate partner violence perpetration, further research on the associations among relationship conflict, intimate partner violence, and COVID-19 exposure and pandemic-specific stress and coping responses, is warranted. It is also crucial to consider how these variables occur among individuals who were part of an underserved population with respect to financial, relational, and mental health related resources prior to the pandemic and during the period of the pandemic characterized by widespread lockdowns. This study is designed to fill that gap.

The focus on the couple relationships of under resourced individuals is essential as the pandemic has not been experienced similarly across various populations. Specifically, individuals from low socioeconomic backgrounds as well as individuals belonging to communities of color have been disproportionately and adversely affected by the pandemic [18]. A national survey completed by United for ALICE indicates communities underneath the poverty threshold are not only faring worse financially, but also physically and emotionally [19]. It appears that an individual’s pre-existing resources, which are social (e.g., supportive friends, family, significant others), financial (e.g., emergency savings, access to food, insurance) and physical (e.g., access to health care, safe shelter) are key to understanding the disparate effects of the pandemic. This tenet is central to the Conservations of Resources (COR) Theory, which has consistently been shown to predict disparate responses to disaster [20]. This theory posits that disasters are less stressful for those with pre-existing resources that can be used to cope with or mitigate the effects of a disaster. Second, everyone is motivated to hold onto whatever resources they have accumulated when the disaster strikes. According to COR, unexpectedly losing resources and/or losing access to resources is a significant source of disaster-related stress. Thus, studying the impact of the pandemic on couple conflict and intimate partner violence (IPV) within an under-resourced population in the United States is an important next step in the application of the COR theory to the unfolding pandemic.

### Hypotheses for the Current Study

First, we hypothesized that the majority of our under-resourced participants would self-report an increase in dyadic conflict during the pandemic, as predicted by COR (H1); conversely, fewer would report that their conflict levels remained the same or even lessened. Second, we hypothesized that individuals who endorsed increased couple conflict during the pandemic would also endorse increased perpetration of and victimization from IPV, greater COVID related exposure and stressors, and more concerning coronavirus coping responses such as increased consumption of alcohol and greater stress and distress (H2). Finally, we expected that reports of intimate partner violence in the dyad would also be related to other negative coronavirus responses and coping strategies (e.g., greater stress, greater alcohol use, reduced sense of well-being) (H3).

## 2. Methods

### 2.1. Community-Academic Partnership

As described in the original publication from these data which focused on individual participants’ COVID-19 exposure and mental health [21], this study emerged from a community-academic partnership that cooperatively launched a regional needs assessment during the summer of 2020. The lead agency, Psychology for All is located in Charlotte, North Carolina. This non-profit organization is dedicated to reducing barriers to accessing mental health services for lower income residents in Central North Carolina. In keeping with community-academic partnership principles, the needs survey utilized in this research was designed to better identify unfolding pandemic exposure, stress responses, and health and mental health-related problems among an under-resourced population. Findings are being used to inform ongoing community actions and program development.

### 2.2. Procedure

After the academic partners obtained Institutional Review Board approval to collect data with an on-line Qualtrics-administered self-report survey, the community partners at Psychology for All distributed a standardized email advertisement to a known primary contact from each of its constituent community partners. All of the constituent community partners had a mission of providing services to ethnic, racial and/or sexual minority groups and/or people experiencing poverty (e.g., C4 Counseling, the Harvest Center, Care Ring, Time Out Youth). Each of these partners was then asked to share the email advertisement, which included study goals, participant risk/benefits, and the survey link, to their respective clientele. Given this participant recruitment strategy, participant response rates could not be determined. Informed consent, which included information about the study aims, procedure, participants rights, potential risks/benefits, remuneration, and investigator contact details, was obtained from all participants prior to survey initiation. All participants also had to confirm that they met the study’s eligibility requirements of (1) being 18 years of age or older, (2) having an annual household income of less than $60,000, and (3) being a resident of Charlotte-Mecklenburg or the surrounding counties (i.e., Gaston, Lincoln, Cabarrus, and Union). Participants also had to report having Medicaid, Medicare or no health insurance. The survey ended immediately for participants who failed to qualify.

Once the survey was launched, measures were presented to each individual in random order to prevent response set effects. In addition, information on a variety of mental health resources were provided routinely throughout the survey (e.g., Psychology for All’s online therapy service application, *Psychology Today*’s national therapist locator services, national crisis phone and text lines). These resources were also offered as part of a written debriefing form which could be printed at survey completion. Finally, participants were given an opportunity to provide an email address from which they could access their preference for a $20.00 Amazon or Walmart e-gift card as remuneration for their effort completing the survey. Recruitment took place during the month of July in 2020. This time period occurred when no vaccines were available, COVID-19 deaths were mounting, and lockdowns were widespread across the North Carolina, US survey area.

### 2.3. Participants

In all, 156 participants were recruited into this convenience sample. However, for the purpose of this study, which focused specifically on couple conflict and intimate partner violence, we excluded participants if they self-reported that they were not a part of a romantic relationship during the pandemic reporting period (*n* = 36). We also eliminated participants whose duration in seconds it took to answer the survey fell more than one standard deviation below the mean response time for the survey (*n* = 10) or if their extended completion time indicated that they were outliers in the data set (*n* = 2). In addition, one additional participant was excluded for failing to answer the question related to their dyadic conflict during the pandemic. Thus, the final sample size for this study constituted 107 participants. However, ns vary slightly across analyses due to missing data.

### 2.4. Measures

The current study focused on a subset of measures from the brief anonymous online survey. Specifically, we analyzed participant demographics, self-reported changes in couple conflict due to the coronavirus, perpetration, and victimization from intimate partner violence during the pandemic period, and how these experiences related to pandemic exposure, stressors and coping responses, stress, and well-being. Given the target population, the agency-to-agency-to participant recruitment strategy, and the online constraints, all constructs were measured with the smallest number of items possible. This reflects our community-academic partnership’s focus on real-world efficiency during the height of the early pandemic.

### 2.5. Demographics

Participants were asked to self-report their age (in categories), education level (less than a bachelor’s degree versus college degree and above), race, ethnicity, self-identified gender, health insurance status (private, Medicaid, Medicare, none), and employment status (full, part-time, unemployed).

### 2.6. Couple Conflict

Consistent with research indicating that it is important to consider changes in couple conflict over time [22] and that conflict is related to IPV perpetration, one item was generated to assess changes in couple conflict during COVID-19. Specifically, participants were asked to indicate on a Likert scale how much their conflict with their partner had changed throughout the pandemic using the following scale: (1) conflict greatly decreased, (2) conflict slightly decreased, (3) no change in conflict, (4) conflict slightly increased and (5) conflict greatly increased. For subsequent analyses and to address hypothesis one, these responses were used to create three groups: conflict slightly or greatly decreased during the pandemic (Conflict Decreased; 43.9%, *n* = 47), conflict stayed the same (No Change, 28%, *n* = 30), and conflict slightly or greatly increased (Conflict Increased, 28%, *n* = 30).

### 2.7. Intimate Partner Violence

Intimate partner violence was measured by two items that were generated from the physical violence subscale from the revised version of the Conflict Tactics Scale (CTS2) [23]. Adequate internal reliability was obtained for this two-item measure (coefficient alpha = 0.67). The first item assessed for the occurrence of mild violence and included these behaviors: ‘threw something that could hurt; twisted an arm or hair; pushed, shoved, grabbed, or slapped’. The second item assessed severe violence: ‘used a knife or gun on; punched or hit with something that could hurt; choked; slammed against a wall; beat up; burned or scalded on purpose; kicked’. For each item, participants were asked to indicate if no one did it, only they did it, only their partner did it, or both of them did it since the start of the pandemic (approximately 6 months). Answers were then collapsed into three non-overlapping groups: No IPV reported (34.6%, *n* = 37); Only Victimization was reported (17.8%, *n* = 19); or Any Perpetration was reported (46.7%, *n* = 50). The perpetrator category, by definition, could also include individuals reporting bi-directional violence (Perpetration and Victimization).

### 2.8. COVID-19 Exposure and Stressors

COVID exposure, illness, and difficulties were measured using eight self-report items that form part of the Coronavirus Stressor Survey (CSS) [24]. The eight items utilized for this study were: ‘became ill due to possible exposure to the Coronavirus’, ‘job requires possible exposure to Coronavirus’, ‘lost job or income due to the Coronavirus pandemic’, ‘had increased responsibilities at home due to the Coronavirus pandemic’, ‘had difficulty getting food, medication or other necessities due to the Coronavirus pandemic’, had difficulty getting needed social support due to the Coronavirus pandemic’, lost health insurance due to the Coronavirus pandemic’, and ‘went on public food assistance due to the Coronavirus pandemic’. Participants were asked to indicate either “yes this happened to me” or “no this didn’t happen to me”. Because the internal consistency for these eight items was low (alpha = 0.51) and to determine whether experiencing particular pandemic events (e.g., becoming ill, increased home responsibilities) were significantly associated with either increased couple conflict or IPV perpetration, these items were considered on an individual basis.

### 2.9. Coronavirus Response Scale-10 (CRS-10)

Pandemic responses, coping efforts and distress levels were measured via the Coronavirus Response Scale-10 (CRS-10) [25]. In the current study, the CRS-10 demonstrated adequate internal reliability (alpha = 0.70). However, the scale’s authors recommend considering each item separately rather than using a total score. Thus, the CRS-10 features ten self-report items that are individually scored based on a five-point Likert scale: (1) ‘a lot less’, (2) ‘less’, (3) ‘about the same’, (4) ‘more’ and (5) ‘a lot more’. Each of the ten items is preceded with the instruction, ‘Due to the Coronavirus’. Higher mean scores indicate greater endorsement of that item.

## 3. Results

The final sample consisted of 107 participants all of whom self-reported being in a romantic relationship during the pandemic. The sample was racially diverse (35% self-identified as white) and fairly evenly split between male (54%) and female (46%) participants. All participants fell under the federal family income level and/or had Medicaid, Medicare or no health insurance. Over half the sample was 36–45 years of age (54%) with participant ages ranging from 18–64 years old. The majority of the sample was not college educated; 82% of participants had an associate degree or less. Prior to analyses, to ensure that the data were normally distributed, the skew and kurtosis of each continuous dependent variable (pandemic responses) was considered. All the variables demonstrated a skew and kurtosis of less than the absolute value of 1 (skew range = 0.01 to 0.41; kurtosis range = 0.00 to −0.98). Given the sample size and these obtained values, the ANOVAs were conducted as planned.

### 3.1. Hypothesis One

We first tested H1 which proposed that the majority of participants would report increased couple conflict during the pandemic. Contrary to expectation, the majority of these participants (*n* = 47, 44%) reported decreased conflict with their partner during the pandemic. We then determined that participants could be separated into three pandemically-related conflict change groups (Conflict Increased, *n* = 30; Conflict Decreased, *n* = 47; and conflict remained the same or No Change, *n* = 30). Chi-square analyses were then conducted to test the associations between conflict group membership and gender, age, education level, employment, and health insurance status. As shown in Table 1, only two variables were found to significantly differ among the three dyadic conflict groups: employment type (full time, part time, or unemployed) and whether the participant had health insurance. Specifically, participants who reported a decrease in conflict were significantly more likely to have health insurance compared to those with no change in conflict or increased conflict (any health insurance 98%, versus 75% and 77%, respectively). Alternately, participants who reported an increase in conflict in their relationships were twice as likely to be unemployed as those with no change in conflict or decreased conflict (24% versus 10% and 13%, respectively).

### 3.2. Differences among the Three Conflict Groups (H2)

As proposed in H2, a Chi-square analysis was conducted to determine if there was a significant association between the three Conflict Change groups and reports of IPV (No IPV, Victimization Only, and Any Perpetration). Contrary to expectation, no significant association was found, *X*^2^(4, *n* = 106) = 1.22, *p* = 0.88. Thirty-five percent of the Conflict Decreased group reported no acts of IPV versus 30% of the No Change group and 40% of the Conflict Increased group.

Given that reported changes in couples conflict were not associated with IPV status, we separately considered differences in COVID-19 exposure, stressors, and responses/impact among the couples conflict groups and the IPV groups. Initially, to determine if significant differences existed among our three groups (Conflict Increased, Conflict Decreased, and No Change in Conflict during the Pandemic) and their reports of eight COVID-19 experiences, a series of Chi-square analyses were conducted. As shown in Table 2, three of eight were significantly associated with differences in couple conflict. Specifically, participants in the Conflict Decreased group were more likely to report that they became ill due to possible exposure to COVID-19 (48.9%) then the No Change participants (20.0%) and the Conflict Increased participants (30.0%), *X*^2^(2, *n* = 107) = 7.25, *p* = 0.027. The other two significant results both indicated that more of the Conflict Increased participants reported experiencing COVID-19 exposure and stressors than the other two couple conflict groups. More people in the Conflict Increased group (63.3%) had difficulty getting needed social support due to the pandemic than in the No Change (36.7%) and Conflict Decreased (27.7%) groups, *X*^2^(2, *n* = 107) = 9.91, *p* = 0.007. Similarly, a greater percent of the Conflict Increased participants (46.7%) lost their health insurance due to the pandemic than the No Change (30.0%) and Conflict Decreased (14.9%) participants, *X*^2^(2, *n* = 107) = 9.24, *p* = 0.01.

As shown in Table 3, a series of one-way ANOVAs revealed that six of the ten items on the pandemically-related response behaviors were endorsed significantly differently among participants from the three couple conflict groups. Among the significant results, two main patterns emerged. In the first pattern, the Increased Conflict group differed significantly from both the Decreased Conflict and the No Change in conflict groups. This was true for participant stress, in which participants in Conflict Increased group (*M* = 3.63, *SD* = 0.89) reported greater increases in COVID-19 stress than participants in Conflict Decreased group (*M* = 2.49, *SD* = 0.75) and in the No Change group (*M* = 2.73, *SD* = 1.05), *F*(2, 104) = 16.03, *p* < 0.001. Similarly, for participants’ emotional distress due to the pandemic, the Conflict Increased group (*M* = 3.50, *SD* = 0.82) reported greater increases than Conflict Decreased group (*M* = 2.62, *SD* = 1.01) and the No Change group (*M* = 2.93, *SD* = 0.83), *F*(2, 104) = 8.60, *p* < 0.001. In the other main pattern, the Conflict Increased group and the Conflict Decreased group only significantly differed from one another. For example, when evaluating participants’ reports of pain due to the pandemic, the Conflict Increased participants (*M* = 3.27, *SD* = 0.94) reported a greater tendency for their pain to increase than did the Conflict Decreased participants (*M* = 2.51, *SD* = 1.10), *F*(2, 104) = 5.10, *p* = 0.008. This was also the case for participants’ fear or worry, as the Conflict Increased group (*M* = 3.53, *SD* = 1.04) reported greater increases in fear or worry due to the virus as compared to the Conflict Decreased group (*M* = 2.70, *SD* = 1.02), *F*(2, 104) = 5.54, *p* = 0.005. The final two significant results had unique patterns. When considering changes in participants’ use of alcohol and/or illicit drugs due to the pandemic, the Conflict Decreased participants (*M* = 2.55, *SD* = 1.19) were significantly more likely to decrease their use compared to both the No Change (*M* = 3.07, *SD* = 1.02) and the Conflict Increased participants (*M* = 3.37, *SD* = 1.00), *F*(2, 104) = 5.41, *p* = 0.006. As shown in Table 3, effect sizes for these significant differences ranged from 0.06 to 0.24.

### 3.3. Differences among the Three IPV Groups (H3)

Chi-square analyses were conducted to see if there was any relationship between IPV group (No IPV, Victimization Only, or Any Perpetration) and the following demographics: gender, age, education level, employment status, and if the participant had health insurance. Only employment type had a significant association with IPV group membership. Specifically, working part time (68.4%) was significantly associated with being a victim of IPV as compared to working full time (31.6%) or being unemployed (0%), *X*^2^(4, *n* = 102) = 25.27, *p* < 0.001. The part time employment rate for those with no IPV was 16.7%.

Chi-square analyses were then conducted to compare IPV group membership and the eight pandemic exposure and stressor experiences. As shown in Table 4, there were only two significant differences among the groups. Substantially more participants who reported no IPV (81.1%) indicated that they had increased responsibilities at home due to the virus compared to participants reporting IPV Victimization Only (26.3%) and participants who reported Any IPV Perpetration (34.0%), *X*^2^(2, *n* = 107) = 23.65, *p* < 0.001. However, fewer participants in the No IPV group (18.9%) reported being likely to go on public food assistance due to the pandemic than in the Victimization Only (47.4%) and the Any Perpetration group (44.0%), *X*^2^(2, *n* = 107) = 7.15, *p* = 0.028.

Lastly, a series of one-way ANOVAs and Tukey’s LSD post hoc tests were conducted to determine if there was a significant association between belonging to one of the three IPV groups and increases or decreases in their coronavirus response behaviors. As reported in Table 5, four significant differences among groups emerged across the 10 response and impact items. First, the No IPV group (*M* = 3.35, *SD* = 0.95) was significantly more likely to report increased stress in response to the pandemic than the Any Perpetration group (*M* = 2.48, *SD* = 0.84), *F*(2, 104) = 9.55, *p* < 0.001. The No IPV group (*M* = 3.22, *SD* = 1.29) also reported being significantly more likely to interact with friends and family due to the pandemic compared participants in the Any Perpetration group (*M* = 2.52, *SD* = 0.84), *F*(2, 104) = 4.27, *p* = 0.017. For emotional distress, the participants in the No IPV group (*M* = 3.22, *SD* = 1.03) recorded greater increases in distress than the Victimization Only participants (*M* = 2.47, *SD* = 0.91), *F*(2, 104) = 3.88, *p* = 0.024. Finally, the No IPV group (*M* = 2.57, *SD* = 1.28) reported using alcohol/illicit drugs significantly less in response to the pandemic as compared to both the Victimization Only group (*M* = 3.26, *SD* = 0.99) and the Any Perpetration IPV group (*M* = 3.06, *SD* = 1.04), *F*(2, 104) = 3.10, *p* = 0.049. As shown in Table 5, the effect sizes for these significant differences were small, ranging from 0.06 to 0.16.

## 4. Discussion

The purpose of this study was to consider how couple conflict and intimate partner violence changed during the pandemic and whether pandemic experiences, responses, and impacts were associated with these changes. This study is unique in that it features a population that was recruited due to their pre-pandemic low levels of resources (lack of income, insurance, and contact with social service agencies) as these individuals were expected to be disproportionately impacted by the pandemic [26]. Moreover, according to the Conservation of Resource Theory [27], individuals experiencing the pandemic from a position of low resources would be expected to be strongly motivated to retain their limited resources, if at all possible. Given that the pandemic challenged access to and retention of social, economic, and physical resources for many across America, the majority of these participants were expected to report increased stress and conflict with their partners. Furthermore, based on a substantial literature base, we hypothesized that those who reported increased couple conflict would also be more likely to report intimate partner violence. Correspondingly, rates of intimate partner violence were expected to be substantial among these participants.

Results were unexpected in that the majority of participants reported decreased couple conflict during the early stages of the pandemic. The remainder of the sample was equally split between participants who reported increased couple conflict and those who reported that the level of conflict in their relationship had not changed. Variables associated with changes in conflict due to the pandemic largely fell into two categories, with some factors differentiating those who reported a decrease in conflict while other factors were highlighted among those reporting an increase in conflict. Specifically, more participants reporting an increase in conflict endorsed having difficulty getting needed social support and losing their health insurance. Consistent with predictions from COR, these individuals were also substantially more likely to be unemployed than those in the No Change or Conflict Decreased Group. Their coronavirus response also differed significantly with greater reports of increased pain, fear and worry, stress, and emotional distress as a result of the virus as compared to participants reporting a decrease in conflict. In most cases, the reports from the No Change group fell in between the other two groups. While these results were obtained using a correlational design, which precludes causal interpretation of these data, as a whole, these findings support the importance of having economic and social resources during disaster in order to mitigate individual and dyadic consequences. These results also provide initial support for the applicability of COR to the pandemic response. Potential health implications related to providing universal access to health insurance and facilitating regular social support, even during a pandemic, are noted.

In some cases, however, findings underscored differences between participants reporting reduced rather than increased conflict; these results might also have public health implications. Specifically, more participants reporting a decrease in conflict were working part time as compared to full-time; none reported being unemployed. Despite their reduced work hours, these participants almost universally had access to health insurance. They also reported a decreased likelihood of using alcohol and/or illicit drugs in an effort to cope with the pandemic as compared to those who reported increased conflict or no change in their relationship conflict due to the pandemic. Previous research has shown that alcohol consumption is associated with feelings of behavioral and mental disengagement, indicating that alcohol-related coping is generally ineffective and could further exacerbate the risk for engaging in intimate partner violence [28,29]. Links among COVID-19 stressors, IPV victimization, substance use, and quarantine behaviors are emerging [30]. Additionally, alcohol consumption is financially costly, and, when consumed in excess, is associated with many other negative physical health and disease processes. Greater universal access to other, less potentially problematic coping strategies (e.g., learning a new skill or hobby such as making music, pet adoption, dancing, cooking, exercise, phone photography), is likely to have public health value. Taken together, these findings suggest that, while most disasters are mitigated some by greater income and employment stability, the pandemic may have some unique characteristics. Health insurance stability is likely to be critical to family functioning during a global health crisis. Furthermore, full time employment might be less advantageous than usual because it necessitates greater contact with others (i.e., increased exposure) and simultaneously offers reduced flexibility to take on increased responsibilities around the home (i.e., home-schooling children, parenting with limited opportunities to go places, increased need to care for sick family members).

It is possible that working part-time during the pandemic provided some of our participants with an essential mix of couple time and economic/insurance benefit. In support of this assertion, a recent study that considered couple functioning among first time mothers during the pandemic also demonstrated that additional couple time in the face of health challenges has a protective function [31]. However, this additional couple time and greater ability to fulfill familial responsibilities may come with a cost in terms of higher risk of intimate partner violence victimization; IPV victimization may spike when interpersonal expectations are raised but not fulfilled to the perpetrators’ satisfaction. Another possible explanation for reduced conflict (but increased risk of IPV victimization) could be that those participants who worked part time were under more financial stress than those who worked full time or those who were unemployed (and were thus eligible for additional benefits, help, and services). Other sources have also found that following the pandemic stimulus payments, IPV decreased [10]. Additional research is needed to determine if these findings replicate and generalize to other geographic regions, types of participants and disasters.

Contrary to expectation and numerous previous pre-pandemic research studies [13,15], there was no significant relationship between increased couple conflict and the self-reported occurrence of intimate partner violence among these participants. However, consistent with previous findings indicating high rates of IPV during the pandemic [10,16,17,32], the majority of these participants indicated that at least one act of intimate partner violence had occurred in their relationship during the pandemic (65.1%). Surprisingly high rates of perpetration were self-reported with 47% of coupled participants indicating that they engaged in at least one act of mild or severe intimate partner violence at some point early on in the pandemic. Of note, consistent with previous studies on the prevalence of bi-directional IPV [33], many within this group reported both perpetration and victimization. These findings highlight the ongoing need to address and prevent this public health problem, even in the midst of a pandemic.

Given that changes in conflict were not associated with reports of intimate partner violence, we considered the relationships between intimate partner violence and COVID-19 exposure, stressors, and responses in a second separate set of analyses. By combining responses to two IPV items, we categorized participants into three groups: Those reporting Any Perpetration; those reporting Victimization Only, and those reporting No IPV during the pandemic period. However, our analyses typically highlighted differences between individuals reporting no IPV in their relationships versus those reporting either victimization only or any perpetration. An interesting picture emerged. Although participants reporting No IPV during the early pandemic were significantly more likely to endorse increased responsibilities at home, higher levels of stress, and increased emotional distress, they also reported consuming less alcohol and/or illicit drugs in response to the pandemic and spending more time with family and friends than those reporting either IPV victimization or perpetration. Greater time spent with family and friends as well as decreased consumption of alcohol and/or drugs may act as crucial protective factors against perpetrating IPV and may also serve to prevent increased conflict among couples and IPV victimization. Finding ways to reduce substance use as a coping strategy while maintaining critical external social support as the pandemic continues to unfold may help promote safety in the home.

### Limitations and Future Directions

Although the population surveyed can be considered a strength as they are often under-represented in research studies and yet they offer an important perspective on the COVID-19 pandemic, limitations related to this sample should be noted. This population was extremely under-resourced and the majority lacked a college education; thus, it is likely that participants in this sample had lower literacy rates compared to the general population [34]. This affects our confidence in the data as all participants were required to complete the survey independently and online. In addition, the survey relied on self-report measures, with some constructs being measured with a single item. Furthermore, the two coronavirus measures utilized in the current study were only recently developed. Neither demonstrated strong internal consistency and thus, each item was considered separately. Advancements in measuring coronavirus exposure, pandemic-specific stressors, and individual’s responses to the pandemic are needed. The sample size was also relatively small and all data were subject to self-report biases. Moreover, because the sample is geographically limited to the Central North Carolina region in the United States, care should be exercised when generalizing these findings to another population, a different geographic region, or even a separate time period in the pandemic. The correlational design also precludes a causal interpretation of these data. Replication of these findings with a larger sample and more robust measurements, including the use of mixed method design, to better understand dyadic dynamics will be essential.

## 5. Conclusions

Nonetheless, this current study adds to the literature in that it considers how an under-resourced population, with high rates of IPV yet considerable reports of reduced couple conflict, experienced and responded to the early stages of the pandemic. Results highlight the continued importance of ensuring access to health insurance, encouraging coping without the consumption of substances, and finding ways to maintain friend and family social support during the pandemic. As previously mentioned, alcohol consumption is significantly linked with ineffective coping outcomes such as mental and behavioral disengagement [28]. More effective interventions for combatting the stressors of the pandemic might also include practicing mindfulness or meditation (e.g., Headspace), decreasing screen time and limiting over-access to news related media, increasing time spent outdoors, facilitating regular exercise, and increasing safe access to a broad base of social support [35,36]. Additionally, adopting an attitude of resilience and fostering a greater tolerance for uncertainty may serve as protective factors for mental health [37]. Further studies should be conducted to consider how at-risk underserved communities continue to cope with the stressors of the pandemic in the face of emerging vaccines, boosters, stoppage of stimulus payments, changing economic conditions, and the appearance of new coronavirus variants such as delta and omicron.

## Figures and Tables

**Table 1 ijerph-19-02608-t001:** Chi-square analyses of the three couple–conflict groups and participants’ demographics.

Demographics	Total Sample *n* = 107	Conflict Decreased *n* = 47	No Change in Conflict *n* = 30	Conflict Increased *n* = 30	Chi-Square (df)	*p*
Name	(%)	(%)	(%)	(%)	*X* ^2^	0.05
GENDER						
Males	57 (53.8)	56.5	46.7	56.7	0.85 (2)	0.65
Females	49 (46.2)	43.5	53.3	43.3		
AGE						
26–35	8 (7.5)	6.4	6.7	10.0	5.92 (4)	0.21
36–45	30 (28.0)	17.0	36.7	36.7		
46+	69 (64.5)	76.6	56.7	53.3		
EDUCATION						
Less than a Bachelor’s Degree	86 (81.9)	77.8	93.3	76.7	3.72 (2)	0.16
Bachelor’s Degree or More	19 (18.1)	22.2	6.7	23.3		
EMPLOYED						
Full Time	45 (43.7)	**26.7**	**58.6**	**55.2**	**14.99** **(4)**	**0.005**
Part Time	42 (40.8)	**60.0**	**31.0**	**20.7**		
Unemployed	16 (15.5)	**13.3**	**10.3**	**24.1**		
HEALTH INSURANCE						
Yes	88 (85.4)	**97.8**	**75.0**	**76.7**	**9.81** **(2)**	**0.007**

Note. Ns varied slightly across analyses due to missing data. Findings that are significant at the *p* < 0.05 level are bolded.

**Table 2 ijerph-19-02608-t002:** Chi-square analyses of the three couple–conflict groups and the Coronavirus Stressor Survey.

COVID-19 Exposure and Stressors: Happened to the Participant	Total Sample *n* = 107	Conflict Decreased *n* = 47	No Change in Conflict *n* = 30	Conflict Increased *n* = 30	Chi-Square	*p*
	%	%	%	%	*X*^2^(df = 2)	0.05
Became Ill from Possible Exposure to the Coronavirus	**38 (35.5)**	**48.9**	**20.0**	**30.0**	**7.25**	**0.027**
Job Requires Possible Exposure to Coronavirus	51 (47.7)	36.2	56.7	56.7	4.44	0.109
Lost Job or Income Due to the Coronavirus Pandemic	52 (48.6)	42.6	50.0	56.7	1.49	0.474
Increased Responsibilities at Home due to the Pandemic	53 (49.5)	42.6	43.3	66.7	4.90	0.086
Difficulty Getting Food, Medication or other Necessities due to the Pandemic	39 (36.4)	31.9	26.7	53.3	5.35	0.069
**Difficulty Getting Needed Social Support due to the Pandemic**	**43 (40.2)**	**27.7**	**36.7**	**63.3**	**9.91**	**0.007**
**Lost Health Insurance due to the Pandemic**	**30 (28)**	**14.9**	**30.0**	**46.7**	**9.24**	**0.010**
Went on Public Food Assistance due to Pandemic	38 (35.5)	34.0	40.0	33.3	0.37	0.831

Note. Findings that are bolded are significant at the *p* < 0.05 level.

**Table 3 ijerph-19-02608-t003:** Comparison of the three couple–conflict groups on the CRS-10 pandemic response items.

CRS-10	Conflict Decreased *n* = 47 Mean (SD)	No Change in Conflict *n* = 30 Mean (SD)	Conflict Increased *n* = 30 Mean (SD)	*F* Value (df = 2)	*p* Value	Effect Size Eta Squared
**My stress is:**	**2.49 ^b^** **(0.75)**	**2.73 ^b^** **(1.05)**	**3.63 ^a^** **(0.89)**	**16.03**	**<0.001**	**0.24**
My interaction with friends and family is:	2.98 (1.13)	2.70 (0.88)	2.77 (1.38)	0.63	0.540	0.01
**My emotional distress is:**	**2.62 ^b^** **(1.01)**	**2.93 ^b^** **(0.83)**	**3.50 ^a^** **(0.82)**	**8.60**	**<0.001**	**0.14**
My physical activity is:	2.83 ^a,b^ (0.96)	3.07 ^b^ (1.17)	2.47 ^a^ (1.04)	2.51	0.086	0.05
**My use of alcohol and/or illicit drugs is:**	**2.55 ^b^** **(1.19)**	**3.07 ^a^** **(1.02)**	**3.37 ^a^** **(1.00)**	**5.41**	**0.006**	**0.09**
**My use of prescription medicine is:**	**2.32 ^b^** **(1.25)**	**2.93 ^a^** **(1.17)**	**2.87 ^a,b^** **(1.17)**	**3.08**	**0.050**	**0.06**
**My pain is:**	**2.51 ^b^** **(1.10)**	**2.97 ^a,b^** **(1.03)**	**3.27 ^a^** **(0.94)**	**5.10**	**0.008**	**0.09**
**My fear or worry is:**	**2.70 ^b^** **(1.02)**	**3.07 ^a,b^** **(1.17)**	**3.53 ^a^** **(1.04)**	**5.54**	**0.005**	**0.10**
My effort to cope with stress is:	3.02 ^b^ (1.03)	3.23 ^a,b^ (1.17)	3.53 ^a^ (0.73)	2.41	0.095	0.04
My overall sense of well-being is:	2.74 (1.03)	3.10 (1.06)	2.73 (0.91)	1.38	0.260	0.03

Note. Responses to each question ranged from 1 to 5 with higher scores indicating greater endorsement during the pandemic. Means that are significantly different from one another using Tukey’s least significant differences post hoc analyses are designated with different superscripts (e.g., a and b). Findings that are significant at the *p* < 0.05 level are bolded.

**Table 4 ijerph-19-02608-t004:** Chi-square analyses of the three IPV groups and the Coronavirus Stressor Survey.

COVID-19 Exposure and Stressors: Happened to the Participant	Total Sample	No IPV *n* = 37 (34.9%)	Victim Only IPV *n* = 19 (17.9%)	Any Perpetration *n* = 50 (47.2%)	Chi-Square	*p*-Value
	*n* (%)	%	%	%	*X*^2^(df = 2)	0.05
Became Ill from Possible Exposure to the Coronavirus	38 (35.5)	37.8	52.6	28.0	3.73	0.155
Job Requires Possible Exposure to Coronavirus	51 (47.7)	45.9	63.2	44.0	2.13	0.345
Lost Job or Income Due to the Coronavirus Pandemic	52 (48.6)	62.2	52.6	38.0	5.09	0.079
**Increased Responsibilities at Home due to the Pandemic**	**53 (49.5)**	**81.1**	**26.3**	**34.0**	**23.65**	**<0.001**
Difficulty Getting Food, Medication or other Necessities due to the Pandemic	39 (36.4)	48.6	26.3	30.0	4.13	0.127
Difficulty Getting Needed Social Support due to the Pandemic	43 (40.2)	54.1	31.6	32.0	4.95	0.080
Lost Health Insurance due to the Pandemic	30 (28)	18.9	31.6	34.0	2.51	0.286
**Went on Public Food Assistance due to Pandemic**	**38 (35.5)**	**18.9**	**47.4**	**44.0**	**7.15**	**0.028**

Note. Findings that are bolded are significant at the *p* < 0.05 level.

**Table 5 ijerph-19-02608-t005:** Comparison of the three IPV groups on the CRS-10 Coronavirus Response Items.

CRS-10	No IPV: *M* (*SD*) *n* = 37 (34.9%)	Victim Only: *M* (*SD*) *n* = 19 (17.9%)	Any Perpetration: *M* (*SD*) *n* = 50 (47.2%)	*F* Value (df = 2)	*p* Value	Effect Size Eta-Squared
**My stress is:**	**3.35 ^a^** **(0.95)**	**2.95 ^a,b^** **(1.08)**	**2.48 ^b^** **(0.84)**	**9.55**	**<0.001**	**0.16**
**My interaction** **w/friends and family is:**	**3.22 ^a^** **(1.29)**	**2.95 ^a,b^** **(1.35)**	**2.52 ^b^** **(0.84)**	**4.27**	**0.017**	**0.08**
**My emotional distress is:**	**3.22 ^a^** **(1.03)**	**2.47 ^b^** **(0.91)**	**2.92 ^a,b^** **(0.90)**	**3.88**	**0.024**	**0.07**
My physical activity is:	2.51 ^a^ (1.22)	2.89 ^a,b^ (0.88)	3.00 ^b^ (0.95)	2.41	0.095	0.05
**My use of alcohol and/or illicit drugs is:**	**2.57 ^a^** **(1.28)**	**3.26 ^b^** **(0.99)**	**3.06 ^b^** **(1.04)**	**3.10**	**0.049**	**0.06**
My use of prescription medicine is:	2.32 ^a^ (1.20)	2.63 ^a,b^ (1.38)	2.92 ^b^ (1.14)	2.59	0.080	0.05
My pain is:	2.65 (1.30)	2.95 (1.18)	2.96 (0.86)	0.97	0.382	0.02
My fear or worry is:	3.05 (1.10)	3.37 (1.17)	2.92 (1.10)	1.12	0.332	0.02
My effort to cope with stress is:	3.32 (0.94)	3.26 (1.05)	3.14 (1.07)	0.36	0.699	0.01
My overall sense of well-being is:	2.84 (0.83)	3.11 (1.05)	2.78 (1.09)	0.74	0.481	0.01

Note. Findings that are bolded are significant at the *p* < 0.05 level. Means that are significantly different from one another using Tukey’s least significant differences post hoc analyses are designated with different superscripts (e.g., a versus b).

## Data Availability

For verification purposes or upon reasonable request, de-identified data can be made available from the first author or the last author of this study.

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
