# Peer review of "Couple Conflict and Intimate Partner Violence during the Early Lockdown of the Pandemic: The Good, the Bad, or Is It Just the Same in a North Carolina, Low-Resource Population?"

_ijerph, 2022, doi:10.3390/ijerph19052608_

Round 1

Reviewer 1 Report

  1. On page 4 (line# 190), what is the internal consistency (Alpha value) of the 2 items from Conflict Tactics Scale (CTS2)?
  2. On page 5 (line# 220), what is the internal consistency (Alpha value) of the 10 items from Coronavirus Response Scale-10 (CRS-10)?
  3. On page 12 (line# 390), the statement “…a decreased likelihood of using alcohol and/or illicit drugs in an effort to cope with the pandemic” will need some explanation or implications.
  4. On page 13 (line# 468), the concluded statement “…encouraging coping without the consumption of alcohol and illegal substances…” can be strengthened by specific intervention addressing the reasons for less consumption of alcohol or drugs.
  5. On page 14 (line#139 to 140), the final statement “…cope with the stressors of the pandemic in the face of vaccines, boosters, stoppage of stimulus payments, changing economic conditions, and the emergence of new variants” will need specific interventions.

Author Response

Reviewer 1:

  1. On page 4 (line# 190), what is the internal consistency (Alpha value) of the 2 items from Conflict Tactics Scale (CTS2)?

We added this information to the manuscript; it was acceptable at .67. 

  1. On page 5 (line# 220), what is the internal consistency (Alpha value) of the 10 items from Coronavirus Response Scale-10 (CRS-10)?

We also added this information to the manuscript even thought this scale is not used as a total score. Again the internal consistency was acceptable (.70). 

  1. On page 12 (line# 390), the statement “…a decreased likelihood of using alcohol and/or illicit drugs in an effort to cope with the pandemic” will need some explanation or implications.

Thank you for this comment. We have added some explanation for this sentence and included direct implications in the discussion secction.

On page 13 (line# 468), the concluded statement “…encouraging coping without the consumption of alcohol and illegal substances…” can be strengthened by specific intervention addressing the reasons for less consumption of alcohol or drugs.

Thank you for this, we have added some specific intervention ideas to the discussion section, as noted above. 

  1. On page 14 (line#139 to 140), the final statement “…cope with the stressors of the pandemic in the face of vaccines, boosters, stoppage of stimulus payments, changing economic conditions, and the emergence of new variants” will need specific interventions.

We added some possible specific interventions to support resilience and to increase tolerance of uncertainty.

Reviewer 2 Report

First of all, I would like to congratulate the authors for this work. For me, this topic is very important and has a lot of value. I have enjoyed reading this manuscript very much.

This paper analyzes changes in intimate partner conflict and intimate violence during the pandemic in low-resource participants.

The manuscript fits well within the scope of the journal and although one cannot claim a good sample size. The methods are well described, but I have some vital recommendations on how to improve them. I will then write recommendations on how to improve it. I will present my recommendation at the end of this paper.

Title

The title is accurate and informative. I think it would be even more informative to readers if the state (North Carolina, USA) was mentioned.

Abstract

The abstract is clear and complies with the general rules for writing a good abstract. However, I would like to see more clearly the main objective of the study in the first few lines of the abstract. I consider this to be the most important section of the paper, as it will be read on many more occasions than even the paper itself. 

Rationale and theoretical framework

As I mentioned, I find this research extremely important. I do not disagree with the authors' justifications and read many very good and current arguments.

Methodology: 

 The methods section is good and well written. However, as I indicated earlier the sample is not important enough although the authors include this as a limitation of the study justifying it adequately. I would also include that this sample pertains to only one region of the United States, so the data should be treated with caution and not generalized. Some recommendations are included below:

- Participants: We suggest including the sample characterization table in this section. However, although it could be interpreted in Table 1, we suggest including a table exclusively for characterization without the inclusion of statistical tests. 
- Instruments. I would like to read more about the validation and psychometric properties of the scales used, specifically the Coronavirus Response Scale 10 CRS. Was confirmatory factor analysis performed on them?
- Statistical analysis: It is recommended to report the justification of the statistical tests used. Did the data follow a normal distribution and was the Kolmogorov-Smirnov test applied to analyze this assumption? It is recommended to clarify this since an ANOVA is used for example, understanding that the data complied with the assumption of normality.

Results: All are displayed correctly and are easy to read for a person not used to interpreting statistical data.

Conclusions. They are clear and give an answer to the stated objectives. Another research that adds to the importance of the treatment of intimate violence to help prevent these situations and to be able to intervene early.

I recommend that this manuscript be sent for another round of review after major revisions, I would be very happy to read this manuscript again.

Author Response

Reviewer 2:

  1. The title is accurate and informative. I think it would be even more informative to readers if the state (North Carolina, USA) was mentioned.

We agreed with the reviewer and accepted this suggestion. The title is changed.

  1. The abstract is clear and complies with the general rules for writing a good abstract. However, I would like to see more clearly the main objective of the study in the first few lines of the abstract. I consider this to be the most important section of the paper, as it will be read on many more occasions than even the paper itself. 

We agree and have added our main objective to the abstract to increase readership and accessibility of our work. 

  1. Participants: We suggest including the sample characterization table in this section. However, although it could be interpreted in Table 1, we suggest including a table exclusively for characterization without the inclusion of statistical tests. 

In view of the space limitations faced by journals and yet in full support of this reviewer’s suggestion, we chose to add a column to the existing table that characterizes the entire sample. 

  1. - Instruments. I would like to read more about the validation and psychometric properties of the scales used, specifically the Coronavirus Response Scale 10 CRS. Was confirmatory factor analysis performed on them?

The authors of this scale recommend using the items separately, rather than as an overall factor or total score, so we did not perform a CFA. However, in light of this reviewer’s comments and in keeping with the first reviewer’s suggestions, we now report the Coefficient Alpha for this scale in the manuscript. It is acceptable (.70)

  1. - Statistical analysis: It is recommended to report the justification of the statistical tests used. Did the data follow a normal distribution and was the Kolmogorov-Smirnov test applied to analyze this assumption? It is recommended to clarify this since an ANOVA is used for example, understanding that the data complied with the assumption of normality.

Thank you for this suggestion. We conducted additional analyses to consider the skew and kurtosis of each of dependent variables. All met the standard assumptions of normality (SKEW + or - the absolute value of one and Kurtosis + or - the absolute value of two). These results are now reported in text in summary form.

Round 2

Reviewer 2 Report

I would like to congratulate the authors for all the improvements made to this new version of the manuscript and for having taken my considerations into account. 

The article has improved significantly with all the revisions and I consider that it should be published.